# Effects of Gum Arabic (*Acacia senegal*) Powder on Characteristics of Carcass and Breast Meat Quality Parameters in Male Broiler Chickens

**DOI:** 10.3390/foods12132526

**Published:** 2023-06-29

**Authors:** Hani H. Al-Baadani, Rashed A. Alhotan, Mahmoud M. Azzam, Gamaleldin M. Suliman, Abdulrahman S. Alharthi, Esam H. Fazea, Ibrahim A. Alhidary

**Affiliations:** Department of Animal Production, College of Food and Agriculture Science, King Saud University, P.O. Box 2460, Riyadh 11451, Saudi Arabia; hsaeed@ksu.edu.sa (H.H.A.-B.); ralhotan@ksu.edu.sa (R.A.A.); mazzam@ksu.edu.sa (M.M.A.); gsuliman@ksu.edu.sa (G.M.S.); abalharthi@ksu.edu.sa (A.S.A.); essamhazaa42@gmail.com (E.H.F.)

**Keywords:** broilers, *Acacia senegal*, carcass, meat quality, fatty acids

## Abstract

Gum Arabic, one of the soluble fibers, is considered a source of natural prebiotics that can be fermented by the activity of the intestinal microbiota and, therefore, may have a positive effect on the performance, carcass characteristics, and meat quality of broilers. Therefore, the objective of this study was to investigate the effects of gum Arabic on performance, carcass characteristics, physicochemical properties, and quality of broiler breast meat. Six dietary treatments (T1–T5) with a basal diet (CON) containing 0.12%, 0.25%, 0.5%, 0.75%, and 1.0% gum Arabic, respectively. A total of 432 one-day-old male broiler chickens were conducted for 35 days. Performance and carcass characteristics, as well as physical, qualitative, and chemical indicators of breast meat, were evaluated. The results showed that the treatments improved overall performance (*p* < 0.05), such as body weight, weight gain, feed conversion ratio, and performance index, except for feed intake, which was lower at T1, T2, and T5 than at CON. Carcass characteristics, body components, and breast meat color or initial pH were not affected by dietary treatments (*p* > 0.05), but ultimate pH (T1 and T2) and color lightness were comparatively lower, and color change (T2–T3) was comparatively higher. Cooking water loss (T2–T4) and shear force (T1–T5) were higher in the treatments. Cohesiveness and resilience were higher in T2, while gumminess and chewiness were lower in T4 and T5. T1 breast meat had higher crude protein content and lower crude fat content, but the other chemical compositions were not affected by the treatments. Some fatty acids in the breast meat were more affected by treatments compared to CON. In conclusion, the present findings showed that gum Arabic (0.12% to 0.75%) has favorable effects on growth performance and some quality indicators of breast meat (such as cooking water loss, shear force, protein, and fat content), especially at the level of 0.12% with no alterations on the carcass characteristics and body components or physical indicators of breast meat.

## 1. Introduction

In recent years, poultry meat production has steadily increased worldwide, mainly due to the continuous improvement of genetic selection, competition among producers, health status, nutrition, and the use of feed additives until the best performance efficiency and meat quality are achieved [1]. Because chicken meat has high nutritional value and low prices compared with other meats, there is more competition for it [2]. Against this background, there are increased efforts to improve meat quality to meet the increasing demand for chicken meat [3]. Feed additives are usually used for preventive purposes to protect chickens from pathogenic infections, while others have a growth-promoting effect [4,5]. Dietary supplementation with antibiotics has been used as a subtherapy to promote broiler chicken health and performance but has been withdrawn due to the establishment of resistant bacteria and high consumer demand for antibiotic-free poultry products [6,7]. The European Union has outlawed the use of antibiotic growth promoters in chicken feed [8]. In response, researchers started looking for the most natural and safest dietary supplements as a result of a considerable decline in the use of antibiotics in chicken feed [9].

Various types of dietary fiber, such as gum Arabic, are considered a source of natural prebiotics for use in broiler diets that deserve attention [10]. Gum Arabic is a soluble dietary fiber found mainly in acacia trees via stem and branch exudates [11] and contains a mixture of complex oligosaccharides and polysaccharides responsible for their prebiotic properties [12,13,14]. Gum Arabic is not degraded by intestinal enzymes, but it is readily fermented by the microbiota when it reaches the cecum and promotes the growth of commensal bacteria [15]. However, promoting the growth of commensal bacteria by soluble fiber such as gum Arabic could improve the health and performance of broiler chickens [16]. Evaluation of meat quality and physicochemical properties is an indicator of fresh chicken meat quality at the point of sale and nutritional value, which determines consumer purchasing power [17,18]. Natural prebiotics (gum Arabic) could play an important role in improving broiler meat quality by increasing feed efficiency and growth due to the change in gut microbial balance [19]. Therefore, dietary supplementation with prebiotics improved carcass characteristics, which may have contributed to improved nutrient utilization in broilers [20]. In another study by Ogbe and Affiku [21], no improvement in meat quality was observed when gum Arabic was administered to broilers.

Although several studies have demonstrated the beneficial effects of gum Arabic on gut health and the performance of broiler chickens [22,23,24], there are few studies on the efficacy of gum Arabic on carcass characteristics and meat quality. Furthermore, based on the previously studied action mechanisms of gum Arabic, it is not degraded in the intestine but fermented by commensal bacteria, which could be reflected in improved overall performance, carcass and meat quality characteristics of broiler chickens, which is the hypothesis based on this current study. Therefore, the objective of this study was to investigate the effects of dietary supplementation with different levels of gum Arabic on performance, carcass characteristics, physicochemical properties, and quality of broiler breast meat.

## 2. Materials and Methods

### 2.1. Preparation of Gum Arabic Powder

Gum Arabic (also known as *Acacia senegal*) was purchased from Abnaa Sayed Elobied Agro Export, Khartoum State, Sudan. It was cleaned and ground into a fine powder used in various levels in broiler feed.

### 2.2. Ethical Approval

The Scientific Research Ethics Committee of King Saud University in Saudi Arabia approved all techniques and methods used in this study (ethics reference number: KSU-SE-20-39).

### 2.3. Housing, Diets, and Management

The current study was conducted on 432 one-day-old male broiler chicks (Ross 308) purchased from a hatchery near Riyadh City. Each chick was separately weighed and randomly assigned to six dietary treatments, each comprising 12 replicates and 6 chicks per replicate. The diet CON was composed of a basal diet without supplement (0.00% gum Arabic), while T1, T2, T3, T4, and T5 were basal diets supplemented with 0.12%, 0.25%, 0.50%, 0.75%, and 1.00% gum Arabic (1.2, 2.5, 5.0, 7.5, and 10.0 g gum Arabic per kg of basal diet, respectively) during the period of 1 to 35 days. The basal diet used in this study for the starter (0–10 days), grower (11–24 days), and finisher (25–35 days) stages was formulated as a mash to meet the total nutrient requirements of broilers (Table 1), as recommended in the Ross 308 Management Guide (Aviagen, 2019, New York, NY, USA). The study was conducted under similar management and optimal environmental conditions according to the Modern Broiler Production Guide in cages with automatic heaters and water dispensers. All broilers were provided with feed and water ad libitum for 35 days. The temperature was maintained at 33 °C for the first week of rearing and then gradually decreased by 3 °C each week until it reached 22 °C for the remainder of the period. Relative humidity in the room chamber was maintained between 65% and 50% (1 to 24 days) and 50% (25 to 35 days). The lighting duration was 24 h during brooding and 23 h during the rest of the period. All broilers were vaccinated against NDV, IBV, and IBDV (Fort Dodge Animal Health, Fort Dodge, IA, USA).

### 2.4. Indicators of Growth Performance

During the study period, body weight and feed intake were recorded, and then average body weight gain (BWG; kg/35 days = difference between final body weight and initial body weight), feed intake (FI; kg/35 days = difference between feed provided weight and residual feed), and feed conversion ratio (FCR; kg/kg = FI/BWG) were calculated according to El-Ratel and colleagues [14]. In addition, the performance index (PI = body weight gain divided by FCR, all multiplied by 100) was calculated throughout the study period [22].

### 2.5. Carcasses and Body Component Variables

At the end of the study (at 35 days of age), 12 birds were randomly selected from each dietary treatment and slaughtered to measure carcass characteristics and body components. After being deprived of feed for 10 h, they were weighed (PSW; pre-slaughter weight) and slaughtered according to animal welfare legislation. The hot carcass (CHW; 15 min after slaughter), cold carcass (CCW; 24 h after slaughter), breast, and legs were weighed. Dressing yield (DY = hot weight/live weight × 100) and weight loss (WL = hot weight − cold weight/hot weight × 100) were calculated [25]. The relative weights of breast and leg meat (breast or leg weight/hot weight × 100) were also calculated [26].

### 2.6. Physical Indicators of Breast Meat

The pH, temperature, and color components of breast meat were measured twice immediately, 15 min, and 24 h after slaughter (initial and ultimate). Breast meat pH and temperature were determined using a pH meter (Hanna Instruments, 211, Woonsocket, RI, USA) as previously described by Khalil and colleagues [25]. Color measurements were taken as duplicates at the initial and final stages after the slaughter of breast meat using a Minolta Chroma Meter (Konica Minolta, CR-400, Tokyo, Japan) with a CIELAB color system for color values (L* = lightness, a* = redness, and b* = yellowing) as previously described by Albrecht and colleagues [27]. The L*, a*, and b* values were used to calculate color change (∆E), chroma (C), hue angle (H), and browning index (BI) according to the methodology of Valizadeh et al. [28] and Cázares-Gallegos and colleagues [29].

### 2.7. Quality of the Breast Meat and Chemical Indicators

Samples were collected from the breast meat to measure water holding capacity (WHC), cooking water loss (CWL), shear force (SF), and texture profile analysis (TPA), according to Adeyemi [30]. WHC% = 100 − [(weight of pre-pressed meat − weight of post-pressed meat)/weight of pre-pressed meat] × 100. CWL % = [(initial weight − cooked weight)/initial weight × 100]. After the cooked breast meat samples, SF and TPA were measured with a TA-HD Texture Analyzer (Stable Micro Systems, Ltd., Godalming, UK). Breast meat chemical composition, such as moisture, crude protein, crude fat, and ash, was analyzed according to the Association of Official Analytical Chemists [31].

### 2.8. Fatty Acid Profile

Fatty acid profiles were analyzed according to the method described by Karadağoğlu et al. [32] and Biesek and colleagues [33]. Briefly, 3 g of dried meat was completely homogenized in a 60 mL mixture of chloroform and methanol (2:1) after filtration (#4 filter paper, >20 to 25 m), then 12 mL of 0.88% KCl solution was added and centrifuged at 2000× *g* for 30 min at room temperature, and the lower solution was transferred to a reaction vial and dried with nitrogen. To the dry samples, 0.5 mL of 0.5 N NaOH in methanol was added, and the samples were heated at 100 °C in a heating block for 5 min and then cooled to room temperature. At the end of this process, total fat (g/100 g muscle) was calculated. 0.5 mL of boron trifluoride in methanol was added to the extract. Then, the extracts were heated to 100 °C for another 5 min and cooled to room temperature. 0.5 mL of hexane and 1 mL of saturated NaCl were added and centrifuged at 1500× *g* for 5 min. The fatty acid methyl esters contained in the upper phase were isolated. HPLC analysis was performed using an Agilent 1260 Series (Palo Alto, CA, USA). Separation was carried out using an Inertsustain AQ-C18 HP column (4.6 mm × 150 mm, 3 μm). The mobile phase consisted of 0.005 N sulfuric acid. The mobile phase was sequentially programmed in a linear gradient for flow rate as follows: 0–4.5 min (0.8 mL/min); 4.5–4.7 min (1 mL/min); 4.7–4.71 min (1 mL/min); 4.71–8.8 (1.2 mL/min); 8.8–9 (1.3 mL/min); 9–23 (1.3 mL/min); and 23–25 (0.8 mL/min). The diode array detector (DAD) was monitored at 210 nm. The injection volume was 5 μL for each of the sample solutions. The column temperature was maintained at 55 °C. Fatty acid peaks were identified using a mixture of standard fatty acids (Larodan Fine Chemicals AB). Fatty acids were calculated based on the peak areas in the chromatogram and expressed as g/100 g of identified fatty acid methyl esters.

### 2.9. Data Analysis

All data were analyzed using SAS software [34] based on a completely randomized design. A statistical comparison between the supplemented diets (T1 to T5) and the basal diet (CON) was performed using Dunnett’s test. Contrasts were also used to determine the linear and quadratic effects of the dietary treatments using regression analysis. Statistical difference statements were considered as *p* < 0.05. The standard error of the means (±SEM) was used to express all values for each parameter.

## 3. Results

### 3.1. Performance Indicators

The effects of dietary treatments on the overall performance of male broilers are shown in Table 2. According to the Dunnett test, BW and BWG are higher when 0.25% to 0.75% gum Arabic (T1 to T4) is fed than CON (*p* < 0.05) during the period from 1 to 35 days of age, whereas FI is lower when T1, T2, and T5 are fed than CON (*p* < 0.05). However, according to the Dunnett test, FCR and PI improve when chickens are fed diets T1 to T5 compared with CON (*p* < 0.05). In addition, contrasts show that increasing the gum Arabic levels (T1 to T5) results in quadratic responses (*p* < 0.05) in BW, BWG, FI, FCR, and PI.

### 3.2. Carcasses and Body Component Variables

The effects of dietary treatments on carcass traits and body composition in male broiler chickens are shown in Table 3. Carcass traits such as PSW, CHW, CCW, and DY and body composition (legs, breast, and back) are not affected by dietary treatments according to Dunnett’s test (*p* > 0.05), nor do they show a linear or quadratic response (*p* > 0.05). On the other hand, the absolute and relative values of WL are higher in T3 compared with CON (*p* < 0.05 by Dunnett’s test). There was also a quadratic response to the dietary treatments on WL (*p* < 0.05).

### 3.3. Physical Indicators of Breast Meat

The effects of dietary treatments on the physical properties of the breast meat of male broiler chickens are shown in Table 4. According to the Dunnett test, carcass temperature is higher in chickens receiving T2 to T5 than in CON (*p* < 0.05) and shows a linear response (*p* < 0.05). Breast meat color and pH values at 15 min post-slaughter are not affected by dietary treatments according to Dunnett’s test (*p* > 0.05), and neither shows a linear or quadratic response (*p* > 0.05). At 24 h after slaughter, pH levels are lower at T1 and T2 compared with CON (*p* < 0.05) and show a quadratic response (*p* < 0.05). Chickens fed T2 and T3 have lower color lightness (L*) and higher color change (∆E) compared with CON (*p* < 0.05 by Dunnett’s test). A quadratic response is also observed (*p* < 0.05). In addition, other breast meat color values and pH differences are not affected by dietary treatments according to Dunnett’s test (*p* > 0.05).

### 3.4. Meat Quality Indicators

The effects of dietary treatments on breast meat quality characteristics of male broiler chickens are shown in Table 5. According to Dunnett’s test, CWL is higher in chicken meat from those fed diets T2 to T4, and SF is higher in chicken meat from those fed diets T1 to T5 than CON (*p* < 0.05), and shows a quadratic response (*p* < 0.05), whereas WHC is not affected by dietary treatments according to Dunnett’s test (*p* > 0.05) and also showed no linear or quadratic response (*p* > 0.05). Cohesiveness and resilience values are higher at T2 than at CON (*p* < 0.05) and show a quadratic response (*p* < 0.05). On the other hand, the values for gumminess and chewiness are lower at T4 and T5 compared with CON (*p* < 0.05), as indicated by Dunnett’s test, and show no linear or quadratic response (*p* > 0.05). Other texture parameters are not affected by dietary treatments according to Dunnett’s test (*p* > 0.05) and also show no linear or quadratic response (*p* > 0.05).

The effects of dietary treatments on the chemical composition of breast meat of male broiler chickens are shown in Table 6. Chickens fed diet T1 had higher crude protein content (CP) and lower crude fat content (CF) (*p* < 0.05), whereas moisture, ash, and organic matter (OM) were not affected by the diet treatments according to Dunnett’s test compared with CON (*p* > 0.05). In addition, the chemical composition of breast meat did not show a linear or quadratic response to dietary treatments (*p* > 0.05).

The effects of dietary treatments on the fatty acid profile of breast meat of male broiler chickens are shown in Table 7. The content of capric acid (C10:0), lauric acid (C12:0), palmitic acid (C16:0), heptadecanoic acid (C17:0), stearic acid (C18:0), pristanic acid (C19:0), and total SF acids is not affected (*p* > 0.05) by the dietary treatments according to Dunnett’s test (*p* > 0.05) and also show no linear or quadratic response (*p* > 0.05). Lignoceric acid (C24:0) content is higher in chickens receiving T2 and lower in chickens receiving T3 (*p* < 0.05). Chickens receiving T2 to T5 have lower myristic acid (C14:0), and T5 have comparatively lower pentadecanoic acid (C15:0) (*p* = 0.0001 and *p* = 0.002, respectively). In addition, there is a quadratic response to C14:0 and a linear response to C24:0 and C15:0 in the dietary treatments (*p* < 0.05). MUFA content of tetradecenoic acid (C14:1 ω5) in T2 and T3 and myristioleic acid (C14:1 ω7) in T2 to T4 is lower compared with CON (*p* < 0.05). In chickens fed T4 and T5, the level of palmitoleic acid (C16:1 ω7) is higher compared to CON (*p* < 0.05). In addition, there is a linear response to the content of C14:1 ω5, C16:1 ω7, and C20:1 ω9 by the dietary treatments (*p* < 0.05). According to the Dunnett test, the content of vaccinic acid (C18:1 ω7), oleic acid (C18:1 ω9), and total MUFA is not affected by the dietary treatments (*p* > 0.05) and show no linear or quadratic response (*p* > 0.05). According to Dunnett’s test, the PUFA content of linoleic acid (C18:2 ω6) and arachidonic acid (C20:4 ω6) is lower in T1 and linolenic acid (C18:3 ω3), omega-3 and omega-6 to omega-3 ratio in T5 (*p* = 0.029, *p* = 0.047, *p* = 0.026, *p* = 0.026, *p* = 0.007, respectively), whereas eicosatrienoic acid (C20:3 ω6) in T1, as well as total PUFA and omega-6 in T2, are higher compared with CON (*p* = 0.028, *p* = 0.035, *p* = 0.037, respectively). In addition, there is a quadratic response to C18:2 ω6, C18:3 ω3, total PUFA, and omega-6 by dietary treatments (*p* < 0.05).

## 4. Discussion

The current results showed a positive effect on the overall performance parameters of male broilers by the addition of gum Arabic during the period of 1 to 35 days, resulting in an increase in BW, BWG (T1 to T4), and PI (T1 to T5), as well as an improvement in FCR (T1 to T5), whereas FI was lower (T1, T2, and T5) compared with CON. These results may indicate that gum Arabic has the ability to ferment through the activity of commensal bacteria in the intestine, reflecting its positive effect on the overall performance parameters of broiler chickens. These results are consistent with those of Siham et al. [22], who studied broiler chickens, and Amber et al. [35], who studied rabbits. Both found that administration of gum Arabic (0.25–0.75%) improved overall performance (BWG and FCR). Similarly, Khan et al. [23] and Al-Baadani et al. [24] concluded that administration of up to 1.00% and 1.50% gum Arabic, respectively, improved overall performance indicators in broilers. In contrast to our findings, gum Arabic did not affect the BWG, FI, or FCR of broilers [36]. Ahmed et al. [37] reported that gum Arabic decreased FI in mice due to increased satiety. However, the physical properties of gum Arabic might contribute to the lower FI in some levels of gum Arabic (T1, T2, and T5) by influencing the rate of passage through the digestive tract and thus improving nutrient utilization, which is reflected in the improved FCR in this study. Adel et al. [15] reported that gum Arabic supplementation has the ability to improve nutrient utilization of broilers. In this study, the increase of PI in chickens fed on gum Arabic may indicate an optimal economic status of production.

Dressing yield and carcass components are important criteria for evaluating carcass characteristics and the slaughter value of broiler chickens [38]. The results of the current study showed that carcass traits, such as PSW, CHW, CCW, and DY, as well as body components (legs, breast, and back), were not affected by dietary treatments. Our results are in agreement with those of Kareem et al. [39] and Rehman et al. [40], who reported that dietary fiber had no effect on carcass and breast yield compared to a basal diet. In contrast, Falaki et al. [41] reported that chickens fed prebiotics had higher carcass and breast meat yields. Dietary supplementation with prebiotics improved carcass characteristics, possibly due to better nutrient utilization by broilers [20]. In addition, our results indicated that WL post-chilling was higher in T3 compared to the control, and this could be assigned to the loss of moisture in the carcass through evaporation due to the lack of body fat. It is well known that carcass fat acts as an insulator preventing the loss of liquids from the carcass. In this study, breast meat quality indicators and physical characteristics, such as WHC, CWL, SF, TPA, pH, temperature, and color components, were evaluated because these variables can significantly affect the quality of fresh chicken meat at the point of sale [18]. In addition, these indicators can determine the consumer’s purchasing power [42]. Although the initial carcass temperature was lower in chickens fed diets T2 to T5 compared with CON, the carcass preparation method could be responsible for this result. The traditional slaughter method used in this study included defeathering in a boiling tank at different temperatures, which is usually associated with rapid processing steps [43]. The pH and color of broiler meat are influenced by many factors, including diet, sex, and pre-slaughter management [44]. In this study, the ultimate pH value (24 h after slaughter) of breast meat was lower in chickens fed diets T1 and T2 compared with a basal diet (CON). However, the results obtained show that the pH values (between 5.8 and 6.1) are within the accepted pH range for broiler meat [45]. The optimal ultimate pH has been reported to be in the range of 6.1 to 5.6 [46]. However, a decline within the normal range of optimal pH may result in pale color and low water-holding capacity [1]. The shelf life of meat is shortened by pH changes due to lipid oxidation or proteolytic degradation during post-slaughter storage [18]. At 24 h after slaughter, chickens fed T2 and T3 diets had lower L* and higher ∆E levels than those fed the basal diet (CON). However, there were no changes in L* and ∆E between the other dietary treatments and the basal diet. At higher a* and b* and lower L*, the color value of the breast meat gradually becomes darker [39]. It seems that the ultimate change in the color value of breast meat is within the normal range at different gum Arabic levels. Schilling et al. [47] demonstrated that pale breast meat has an L* value greater than 50. Matarneh et al. [48] reported that breast meat color could be influenced by the treatment of chickens and carcasses before and after slaughter, which showed a positive correlation with a decrease in pH value post-slaughter (initial and ultimate). Changes in meat color from the normal (light red) state are usually the first signs of a decline in nutrient content and quality [49]. In addition, heme pigments, moisture content, strain, and physical condition of the protein can contribute to variations in meat color [50].

According to Valenta et al. [51], breast meat is the most valuable and commercially important component of chicken carcasses. In this study, CWL was higher in chicken meat in T2 to T4 and SF in chicken meat in T1 to T5, while WHC was not affected by dietary treatments compared with CON. In addition, cohesion and elasticity scores were higher for breast meat in T2, while gumminess and chewiness scores were lower for breast meat in T4 and T5 compared with CON. These results are in agreement with those of Khalafalla et al. [52], who indicated that dietary fibers such as prebiotics do not affect water loss from broiler breast meat during cooking. Water-holding capacity (WHC) is one of the most important functional markers of meat [53]. If the ability to retain water is weak, the meat will not be juicy [54]. In addition, CWL is an indicator of water loss during cooking due to shrinkage and, therefore, may be directly correlated with loss of juiciness [55]. Shear force (SF) was higher in breast meat when chickens were fed gum Arabic, which could be due to the numerical increase in breast weight and the associated larger muscle fibers [56]. Texture profile analysis (TPA), such as hardness, springiness, cohesiveness, gumminess, chewiness, and resilience, is an indicator of the sensory and quality characteristics of chicken breast meat that mimic mechanical digestion in humans and are thus related to overall consumer satisfaction with meat quality [17,57].

Breast meat of male broilers fed T1 (0.12% gum Arabic) had high crude protein content and low crude fat content compared with CON. This suggests that the use of gum Arabic as a source of dietary fiber could improve protein metabolism and reduce the deposition of fat in broilers’ breast meat. Ahmed et al. [11] reported that gum Arabic is capable of repressing fat accumulation by altering the expression of fat metabolism genes. Moreover, gum Arabic was shown to have a fat-lowering effect through different suggested mechanisms [58]. Therefore, the decrease in crude fat in the meat of the treated group could be directly attributed to the level of 0.12% gum Arabic (T1). Abdelwahid et al. [59] stated that a diet supplemented with prebiotics (0.2% manno-oligosaccharides) could favorably improve the protein and fat content of broiler breast meat. On the other hand, El-Ratel et al. [14] found that gum Arabic (0.1% and 0.2%) did not improve the chemical analysis of rabbit meat. The fatty acid profile of chicken meat is an important indicator of meat quality. Because customers are interested in the nutritional value of meat, we focused on breast meat for fatty acid analysis in our study. The fatty acid profile of meat has a significant impact on broiler meat quality and human health [60]. The current study shows that dietary treatments can affect the content of some fatty acids, such as lignoceric acid, myristic acid, pentadecanoic acid, tetradecenoic acid, myristioleic acid, palmitoleic acid, linoleic acid, arachidonic acid, eicosatrienoic acid, and the total content of PUFA and omega-6 and omega-3 fatty acids in broiler chickens. Total PUFA content was higher in T2 (33.16%) and lower in T5 (24.44%), but T5 did not differ when compared to CON; thus, the inclusion of gum Arabic at a level of 0.25% (T2) increased in omega-6 concentration, while it decreased in omega-3 concentration with high levels of gum Arabic (T5). A study by Mattioli et al. [61] showed that the addition of prebiotics to the diet did not increase the fatty acid content of rabbit meat. However, there is a need to confirm the mechanism of action of gum Arabic by conducting future studies on the characteristics of breast meat quality and the fatty acid profile of broiler chickens.

## 5. Conclusions

We concluded that gum Arabic (0.12% to 0.75%) could be an effective strategy to improve growth performance indicators while it has no alterations on carcass characteristics and body components or physical indicators of breast meat. Dietary supplements with gum Arabic, especially at the level of 0.12%, improved breast meat quality indicators such as cooking water loss and shear force as well as breast meat chemical composition through increased protein content and reduced the deposit of fat. Therefore, further studies are needed to clarify the role of gum Arabic in carcass characteristics, physical properties, and breast meat quality of broiler chickens.

## Figures and Tables

**Table 1 foods-12-02526-t001:** Feed composition and nutritive value of basal diet.

Ingredient	Basal Diet (g/kg)		
	Starter	Grower	Finisher
Corn	527	574	614
Soybean meal (CP 48%)	390	340	291
Corn oil	37	44	53
Dicalcium phosphate	18	16	15
Ground limestone	10	9.2	8.6
Salt	4.2	3.2	3.3
DL-Methionine (88%)	3.5	3.2	2.9
L-Lysine HCl (78.4%)	2.0	1.9	1.9
L-Threonine (98.5%)	1.3	1.1	0.9
Vitamin premix ^a^	2.5	2.5	2.5
Mineral premix ^b^	2.5	2.5	2.5
Choline CL (60%)	0.9	0.9	1.0
Sodium bicarbonate	0.1	1.5	3.4
Total	1000	1000	1000
Content in 1 kg of diet (calculated nutrient)			
Metabolizable energy, kcal	3000	3100	3200
Crude protein, g	232.9	211.5	190.9
Crude fat, g	65.1	72.6	81.6
Crude fiber, g	28.3	27.2	26.1
Calcium, g	9.6	8.7	7.9
Non-phytate P, g	4.8	4.4	4.0
Dig. lysine, g	12.8	11.5	10.3
Dig. methionine + cysteine, g	9.5	8.7	8.0
Dig. threonine, g	8.6	7.7	6.9
Dig. arginine, g	14.3	12.8	11.4

^a^ Containing by kg of vitamin premix: Vit. A 2,400,000 IU; Vit. D 1,000,000 IU; Vit. E 16,000 IU; Vit. K 800 mg; Vit. B1 600 mg; Vit. B2 1600 mg; Vit. B6 1000 mg; Vit. B12 6 mg; Biotin 40 mg; Folic Acid 400 mg; Niacin 8000 mg; and Pantothenic Acid 3000 mg. ^b^ Containing by kg of mineral premix: Cobalt 80 mg; Copper 2000 mg; Iodine 400 mg; Iron 1200 mg; Manganese 18,000 mg; Selenium 60 mg; and Zinc 14,000 mg.

**Table 2 foods-12-02526-t002:** Effect of dietary treatments on overall growth performance of broiler chickens.

Parameters ^1^	Dietary Treatments (TR) ^2^						SEM ^3^	*p*-Value ^4^		
	CON	T1	T2	T3	T4	T5		TR	L	Q
BW, g/1 d	43.98	43.97	43.97	44.00	43.88	43.86	0.04	0.156	0.332	0.218
BW, g/35 d	2222 ^b^	2424 ^a^	2383 ^a^	2387 ^a^	2378 ^a^	2332 ^b^	32.9	0.001	<0.0001	0.007
BWG, g/35 d	2178 ^b^	2380 ^a^	2339 ^a^	2343 ^a^	2334 ^a^	2288 ^b^	30.1	0.001	<0.0001	0.006
FI, g/35 d	3281 ^a^	3085 ^b^	3091 ^b^	3162 ^a^	3166 ^a^	3128 ^b^	33.9	0.004	0.0003	0.029
FCR, g/g	1.51 ^a^	1.30 ^b^	1.32 ^b^	1.35 ^b^	1.36 ^b^	1.37 ^b^	0.01	<0.0001	<0.0001	<0.0001
PI	145 ^b^	184 ^a^	177 ^a^	174 ^a^	172 ^a^	168 ^a^	3.71	<0.0001	<0.0001	<0.0001

^a,b^ When means for each parameter within a row do not have a common superscript letter compared with the basal diet (CON), they differ significantly according to Dunnett’s test (*p* < 0.05). ^1^ BW = body weight; BWG = body weight gain; FI = feed intake; FCR = feed conversion ratio; and PI = performance index. ^2^ Dietary treatments (CON, T1, T2, T3, T4, and T5) were supplemented with 0.00%, 0.12%, 0.25%, 0.50%, 0.75%, and 1.00% gum Arabic. ^3^ SEM = standard error of means for diets effect. ^4^ TR = dietary treatments response; L = linear response; and Q = quadratic response.

**Table 3 foods-12-02526-t003:** Effect of dietary treatments on carcass properties and body components of broiler chickens.

Parameters ^1^	Dietary Treatments (TR) ^2^						SEM ^3^	*p*-Value ^4^		
	CON	T1	T2	T3	T4	T5		TR	L	Q
Carcass properties										
PSW, kg	2.25	2.40	2.32	2.45	2.43	2.45	0.08	0.453	0.084	0.547
CHW, kg	1.48	1.59	1.53	1.60	1.59	1.59	0.06	0.654	0.130	0.496
CCW, kg	1.43	1.53	1.47	1.54	1.52	1.53	0.06	0.680	0.144	0.546
WL, g	52.1 ^b^	54.1 ^b^	57.9 ^b^	63.4 ^a^	58.5 ^b^	57.5 ^b^	2.46	0.040	0.025	0.031
WL, %	3.53 ^b^	3.41 ^b^	3.81 ^b^	3.95 ^a^	3.71 ^b^	3.63 ^b^	0.12	0.048	0.225	0.051
DY, %	65.6	66.2	65.7	65.4	65.2	64.9	0.57	0.688	0.848	0.532
Body components										
Legs, g	556	582	571	611	591	591	21.9	0.597	0.173	0.417
Legs, %	37.61	36.68	37.39	38.03	37.26	37.24	0.59	0.724	0.661	0.795
Breast, g	573	646	606	629	644	646	31.6	0.515	0.085	0.647
Breast, %	38.56	40.63	39.42	39.20	40.47	40.52	0.91	0.501	0.144	0.983
Back, g	123	116	115	121	119	115	6.3	0.912	0.406	0.863
Back, %	8.30	7.32	7.49	7.56	7.53	7.26	0.35	0.369	0.030	0.400

^a,b^ When means for each parameter within a row do not have a common superscript letter compared with the basal diet (CON), they differ significantly according to Dunnett’s test (*p* < 0.05). ^1^ PSW = pre-slaughter weight; CHW = carcass hot weight; CCW= carcass cold weight; WL = weight loss; DY = dressing yield; and A. fat = Abdominal fat. Body components were computed as a ratio to the carcass weight. ^2^ Dietary treatments (CON, T1, T2, T3, T4, and T5) were supplemented with 0.00%, 0.12%, 0.25%, 0.50%, 0.75%, and 1.00% gum Arabic. ^3^ SEM = Standard error of means for diets effect. ^4^ TR = dietary treatments response; L = linear response; and Q = quadratic response.

**Table 4 foods-12-02526-t004:** Effect of dietary treatments on breast meat physical properties of broiler chickens.

Parameters ^1^	Dietary Treatments (TR) ^2^						SEM ^3^	*p*-Value ^4^		
	CON	T1	T2	T3	T4	T5		TR	L	Q
At 15 min post-slaughter (initial)										
Temp., °C	32.5 ^a^	31.9 ^a^	29.5 ^b^	29.4 ^b^	29.8 ^b^	28.2 ^b^	0.37	<0.0001	<0.0001	0.085
pH*i*	6.2	6.2	6.2	6.2	6.2	6.1	0.03	0.336	0.060	0.302
Color values										
L*	47.9	47.7	47.6	47.0	48.83	46.71	0.66	0.315	0.673	0.740
a*	0.34	0.61	0.78	1.36	0.95	1.85	0.50	0.315	0.149	0.889
b*	24.8	20.7	22.6	24.8	21.46	24.78	0.51	0.162	0.019	0.300
∆E	46.7	46.7	46.9	47.5	45.70	47.86	0.67	0.303	0.805	0.664
C*	10.5	8.7	9.5	9.8	9.22	9.51	0.51	0.204	0.032	0.301
H*	34.6	−21.0	30.9	64.0	47.13	61.35	22.6	0.111	0.939	0.729
BI	24.8	20.7	22.6	24.8	21.46	24.78	1.71	0.324	0.296	0.310
At 24-h post-slaughter (ultimate)										
pH*u*	6.0 ^a^	5.8 ^b^	5.8 ^b^	6.0 ^a^	6.0 ^a^	6.1 ^a^	0.02	0.001	0.045	<0.0001
pHd	0.21	0.26	0.23	0.25	0.21	0.12	0.03	0.078	0.619	0.054
Color values										
L*	47.8 ^a^	44.9 ^a^	43.3 ^b^	43.6 ^b^	48.1 ^a^	46.6 ^a^	0.87	0.003	0.013	0.0007
a*	2.74	2.91	4.90	4.62	3.95	3.74	0.69	0.181	0.095	0.052
b*	16.2	13.3	13.5	14.7	14.4	14.6	0.71	0.077	0.008	0.044
∆E	48.2 ^b^	50.3 ^b^	52.1 ^a^	51.9 ^a^	47.5 ^b^	49.0 ^b^	0.84	0.005	0.036	0.001
C*	16.6	13.7	14.7	15.4	15.0	15.2	0.70	0.117	0.023	0.137
H*	80.2	77.6	70.1	72.7	73.9	76.0	2.71	0.132	0.043	0.020
BI	45.2	39.3	45.4	48.3	41.1	43.3	2.45	0.143	0.532	0.584

^a,b^ When means for each parameter within a row do not have a common superscript letter compared with the basal diet (CON), they differ significantly according to Dunnett’s test (*p* < 0.05). ^1^ L = lightness; a = redness; b = yellowness; pHd = pH different; ∆E = color change; C = Chroma1; H = hue angle; and BI = browning index. ^2^ Dietary treatments (CON, T1, T2, T3, T4, and T5) were supplemented with 0.00%, 0.12%, 0.25%, 0.50%, 0.75%, and 1.00% gum Arabic. ^3^ SEM = standard error of means for diets effect. ^4^ TR = dietary treatments response; L = linear response; and Q = quadratic response.

**Table 5 foods-12-02526-t005:** Effect of dietary treatments on breast meat quality properties of broiler chickens.

Parameter ^1^	Dietary Treatments (TR) ^2^						SEM ^3^	*p*-Value ^4^		
	CON	T1	T2	T3	T4	T5		TR	L	Q
WHC%	28.73	25.80	28.71	27.73	31.18	28.68	1.13	0.054	0.802	0.693
CWL%	9.55 ^b^	5.82 ^b^	17.31 ^a^	21.97 ^a^	14.74 ^a^	7.67 ^b^	1.63	<0.0001	0.031	<0.0001
SF kgf	6.61 ^b^	8.71 ^a^	10.59 ^a^	9.62 ^a^	9.28 ^a^	10.46 ^a^	0.44	<0.0001	<0.0001	0.003
Texture profile analysis (TPA)										
Hardness	13.99	13.38	13.79	12.19	10.58	11.10	0.99	0.080	0.108	0.790
Springiness	0.64	0.64	0.67	0.63	0.61	0.61	0.01	0.063	0.714	0.145
Cohesiveness	0.36 ^b^	0.37 ^b^	0.43 ^a^	0.39 ^b^	0.36 ^b^	0.35 ^b^	0.01	0.003	0.180	0.001
Gumminess	5.16 ^a^	5.21 ^a^	5.98 ^a^	4.98 ^a^	3.93 ^b^	4.02 ^b^	0.46	0.023	0.504	0.105
Chewiness	3.29 ^a^	3.41 ^a^	3.98 ^a^	3.16 ^a^	2.47 ^b^	2.55 ^b^	0.32	0.017	0.628	0.086
Resilience	0.15 ^b^	0.16 ^b^	0.18 ^a^	0.17 ^b^	0.15 ^b^	0.16 ^b^	0.01	0.017	0.170	0.012

^a,b^ When means for each parameter within a row do not have a common superscript letter compared with the basal diet (CON), they differ significantly according to Dunnett’s test (*p* < 0.05). ^1^ WHC = water-holding capacity; CWL = cooking water loss; and SF = shear force. ^2^ Dietary treatments (CON, T1, T2, T3, T4, and T5) were supplemented with 0.00%, 0.12%, 0.25%, 0.50%, 0.75%, and 1.00% gum Arabic. ^3^ SEM = standard error of means for diets effect. ^4^ TR = dietary treatments response; L = linear response; and Q = quadratic response.

**Table 6 foods-12-02526-t006:** Effect of dietary treatments on breast meat chemical composition of broiler chickens.

Parameters ^1^	Dietary Treatments (TR) ^2^						SEM ^3^	*p*-Value ^4^		
	CON	T1	T2	T3	T4	T5		TR	L	Q
Moisture, %	73.08	73.07	73.64	73.08	73.80	73.19	0.34	0.498	0.476	0.462
CP, %	22.98 ^b^	24.96 ^a^	22.53 ^b^	23.75 ^b^	23.05 ^b^	23.44 ^b^	0.42	0.008	0.231	0.830
CF, %	2.59 ^a^	0.75 ^b^	2.50 ^a^	1.98 ^a^	1.90 ^a^	2.03 ^a^	0.39	0.039	0.089	0.551
Ash, %	1.33	1.22	1.32	1.18	1.24	1.33	0.08	0.650	0.409	0.239
OM, %	98.67	98.78	98.68	98.82	96.96	98.67	0.73	0.462	0.727	0.895

^a,b^ When means for each parameter within a row do not have a common superscript letter compared with the basal diet (CON), they differ significantly according to Dunnett’s test (*p* < 0.05). ^1^ CP = crude protein; CF = crude fat; and OM = organic matter (100-Ash). ^2^ Dietary treatments (CON, T1, T2, T3, T4, and T5) were supplemented with 0.00%, 0.12%, 0.25%, 0.50%, 0.75%, and 1.00% gum Arabic. ^3^ SEM = standard error of means for diets effect. ^4^ TR = dietary treatments response; L = linear response; and Q = quadratic response.

**Table 7 foods-12-02526-t007:** Effect of dietary treatments on breast meat fatty acid profile of broiler chickens.

Parameters	Dietary Treatments (TR) ^1^						SEM ^2^	*p*-Value ^3^		
	CON	T1	T2	T3	T4	T5		TR	L	Q
Saturated fatty acids (SFA)										
C10:0	0.24	0.29	0.22	0.18	0.20	0.17	0.03	0.107	0.335	0.599
C12:0	0.36	0.43	0.41	0.42	0.42	0.22	0.10	0.757	0.880	0.193
C14:0	3.11 ^a^	2.35 ^a^	1.23 ^b^	0.92 ^b^	1.46 ^b^	0.77 ^b^	0.27	<0.0001	<0.0001	0.017
C15:0	1.16 ^a^	1.45 ^a^	0.74 ^a^	0.67 ^a^	0.72 ^a^	0.56 ^b^	0.13	0.002	0.040	0.769
C16:0	20.68	18.46	20.54	16.15	21.09	23.67	2.03	0.219	0.757	0.072
C17:0	0.27	1.04	0.46	0.48	0.42	0.25	0.19	0.118	0.263	0.208
C18:0	6.24	5.80	7.21	6.55	6.24	7.23	0.38	0.088	0.392	0.940
C19:0	0.23	0.29	0.29	0.17	0.20	0.28	0.04	0.206	0.695	0.706
C24:0	4.91 ^b^	4.41 ^b^	5.26 ^a^	0.21 ^c^	4.52 ^b^	-	0.08	<0.0001	0.001	0.495
ΣSFA	35.95	33.44	34.57	25.30	34.01	32.97	2.54	0.881	0.180	0.155
Monounsaturated fatty acids (MUFA)										
C14:1 ω5	0.73 ^a^	0.78 ^a^	0.37 ^b^	0.40 ^b^	0.44 ^a^	0.29 ^b^	0.07	0.003	0.005	0.303
C14:1 ω7	1.36 ^a^	0.99 ^a^	0.41 ^b^	0.46 ^b^	0.41 ^b^	-	0.23	0.045	0.267	0.174
C16:1 ω7	2.08 ^b^	2.31 ^b^	1.99 ^b^	2.89 ^b^	3.36 ^a^	3.77 ^a^	0.25	0.0005	0.023	0.114
C18:1 ω7	1.52	1.46	1.41	1.56	1.54	1.69	0.07	0.129	0.872	0.071
C18:1 ω9	33.33	34.50	32.03	34.78	33.47	36.70	1.64	0.485	0.600	0.325
C20:1 ω9	0.19 ^b^	0.32 ^a^	0.24 ^b^	0.26 ^b^	0.23 ^b^	0.25 ^b^	0.02	0.035	0.022	0.224
ΣMUFA	38.71	40.17	36.35	40.04	39.46	42.73	1.58	0.179	0.554	0.151
Polyunsaturated fatty acids (PUFA)										
C18:2 ω6	25.71 ^a^	23.62 ^b^	27.62 ^a^	26.37 ^a^	25.32 ^a^	21.00 ^b^	1.30	0.029	0.523	0.022
C18:3 ω3	0.72 ^a^	0.74 ^a^	0.71 ^a^	0.76 ^a^	0.72 ^a^	0.42 ^b^	0.07	0.026	0.543	0.019
C18:3 ω6	0.18	0.23	0.19	0.20	0.22	0.31	0.03	0.121	0.199	0.147
C20:2 ω6	0.40	0.38	0.52	0.42	0.41	0.43	0.06	0.593	0.641	0.454
C20:3 ω6	0.36 ^b^	1.59 ^a^	0.43 ^b^	0.28 ^b^	0.35 ^b^	0.32 ^b^	0.32	0.028	0.500	0.807
C20:4 ω6	2.04 ^a^	1.46 ^b^	2.60 ^a^	1.79 ^a^	1.83 ^a^	0.95 ^b^	0.33	0.047	0.396	0.088
C22:4 ω6	0.61	0.85	0.78	0.65	0.59	0.50	0.08	0.081	0.531	0.074
ΣPUFA	29.68 ^b^	29.30 ^b^	33.16 ^a^	30.67 ^b^	29.59 ^b^	24.44 ^b^	1.62	0.035	0.890	0.007
Σ ω6	28.97 ^b^	28.56 ^b^	32.45 ^a^	29.91 ^b^	28.87 ^b^	24.03 ^b^	1.57	0.037	0.907	0.007
Σ ω3	0.72 ^a^	0.74 ^a^	0.71 ^a^	0.76 ^a^	0.72 ^a^	0.42 ^b^	0.07	0.026	0.543	0.019
Σ ω6/Σ ω3	40.44 ^b^	38.44 ^b^	46.99 ^b^	41.67 ^b^	40.18 ^b^	64.48 ^a^	4.62	0.007	0.258	0.067
ΣPUFA/ΣSFA	0.83	0.88	0.96	1.49	0.88	0.76	0.21	0.202	0.480	0.079

^a,b^ When means for each parameter within a row do not have a common superscript letter compared with the basal diet (CON), they differ significantly according to Dunnett’s test (*p* < 0.05). ^1^ Dietary treatments (CON, T1, T2, T3, T4, and T5) were supplemented with 0.00%, 0.12%, 0.25%, 0.50%, 0.75%, and 1.00% gum Arabic. ^2^ SEM = standard error of means for diets effect. ^3^ TR = dietary treatments response; L = linear response; and Q = quadratic response.

## Data Availability

The data and analyses presented in this paper are freely available in the thesis by the first author (H.H.A.-B.).

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
