# Peer review of "Effects of Gum Arabic (Acacia senegal) Powder on Characteristics of Carcass and Breast Meat Quality Parameters in Male Broiler Chickens"

_foods, 2023, doi:10.3390/foods12132526_

Round 1

Reviewer 1 Report

The total number of chicks used for the study is relatively small, but sufficient to draw sound conclusions on the subject. However, the major shortcoming of the Material and methods is the very wide range between the lightest and the heaviest chicks (44±17 g), which were randomly allocated to each treatment. Accordingly, the individual weights of the chicks in each treatment/replication probably varied greatly and could therefore significantly influence the final results.

To obtain scientifically sound conclusions, the average weight of the chicks in each treatment and each replicate should be known at the beginning of the experiment so that it is uniform (consisting of the proportional number of lighter and heavier chicks). Otherwise, the initial weight distribution of the chicks in each treatment is unknown. So if the authors can provide this data (the average weight of chicks for control and T1 – T5 groups at the beginning of the experiment), I am willing to reconsider the evaluation of this work.

Additionally, according to ROS 308 broiler performance objectives, FCR at day 35 is 1.548 and FI 3.319 g. Could you please explain how it is possible that group T1 (with a negligible percentage of gum Arabic, which is not the case in the papers you cited in Discussion) had the best values for FCR (1.30) and for FI (3.08 g) in your study?

Comments by rows

176-177: this statement is partially incorrect; according to the data presented in Table 2. PI was improved in all the treatments, not only from T2 to T5.

196: Table 3. I assume that the weights of the body components are in grams (g). "gm" is not correct.

Author Response

Date: 20, June, 2022‎

Dear Dr. Reviewer #1‎, ‎

On behalf of my colleagues, we would like to thank you for considering our ‎‎manuscript entitled "Effects of Gum Arabic (Acacia Senegal) powder on Characteristics of Carcass, Breast ‎Meat ‎Physical and Quality in Male Broiler Chickens". ‎

We have revised the manuscript based on the reviewers’ comments point-by point ‎and ‎our responses to the reviewer are marked up using the “Track Changes” each ‎specific ‎comment.

We hope the changes made are satisfactory to Your Excellency and to the ‎respected ‎reviewers. ‎

Best regards! ‎

Prof. Ibrahim A. Alhidary

Q1: The total number of chicks used for the study is relatively small, but sufficient to draw sound conclusions on the subject. However, the major shortcoming of the Material and methods is the very wide range between the lightest and the heaviest chicks (44±17 g), which were randomly allocated to each treatment. Accordingly, the individual weights of the chicks in each treatment/replication probably varied greatly and could therefore significantly influence the final results.

Authors’ Response: Thank you for your notes. Indeed, there were unintended errors (between maximum and minimum = 0.17g but "±17 g" is not correct), so we add all body weights of the chicks at‎ 1 and 35 days in table 2 and remove (44±17 g) in line 92.

Q2: To obtain scientifically sound conclusions, the average weight of the chicks in each treatment and each replicate should be known at the beginning of the experiment so that it is uniform (consisting of the proportional number of lighter and heavier chicks). Otherwise, the initial weight distribution of the chicks in each treatment is unknown. So if the authors can provide this data (the average weight of chicks for control and T1 – T5 groups at the beginning of the experiment), I am willing to reconsider the evaluation of this work.

Authors’ Response: Once again, thank you very much for your interest in this main point. Done as requested in table 2.

Q3: Additionally, according to ROS 308 broiler performance objectives, FCR at day 35 is 1.548 and FI 3.319 g. Could you please explain how it is possible that group T1 (with a negligible percentage of gum Arabic, which is not the case in the papers you cited in Discussion) had the best values for FCR (1.30) and for FI (3.08 g) in your study?

Authors’ Response: Thank you for your notes. we have referred in lines 330-334.

Actually, we look that FCR was improved in chickens fed on 0.12 to 1.0% gum Arabic levels than CON. but could be due to the physical and chemical properties of gum Arabic as 0.12%, result that positive effect on FCR.

Therefore, consequent improved in FCR and changes in feed intake is also expected.

There is a need to confirm the mechanism of action of gum ‎Arabic by conducting future studies thus We are continuing to conduct studies on investigating the mechanism of action of gum arabic in broiler chickens.

Q4: 176-177: this statement is partially incorrect; according to the data presented in Table 2. PI was improved in all the treatments, not only from T2 to T5.

Authors’ Response: Thank you for your comments. ‎ Done as requested in line 196.

Q5: 196: Table 3. I assume that the weights of the body components are in grams (g). "gm" is not correct.

Authors’ Response: Authors’ Response: Thanks for your feedback. Done as requested in Table 3.

The manuscript has been completely revised, with some language changes made and ‎improved from our point of view, on the other hand, I appreciate your efforts in your ‎valuable comments and question, which gave me the opportunity to improve the ‎throughout manuscript.

Please if there are any opinions, guide us to correct it. ‎

satisfactory to you ‎

Thanks so much for your efforts. Your feedbacks are very valuable and will improve ‎my research skills and biological insight on my future studies.

--

Prof. Ibrahim A. Alhidary

Reviewer 2 Report

The current manusript entitled "Effects of Gum Arabic (Acacia Senegal) powder on Characteristics of Carcass, Breast Meat Physical and Quality in Male  Broiler Chickens" is about the effect of using gum Arabic in the broiler nutrition. Despite being interesting in the subject, however, it needs improvements in the different sections, especially discussion. The authors need to elaborate on their results and mention about the mechanisms. My specific comments are in the pdf file. The similarity is also 33% which should be less than 20%.

Moderate improvement is needed. 

Author Response

Date: 20, June, 2022‎

Dear Dr. Reviewer #2‎, ‎

On behalf of my colleagues, we would like to thank you for considering our ‎‎manuscript entitled "Effects of Gum Arabic (Acacia Senegal) powder on Characteristics of Carcass, Breast ‎Meat ‎Physical and Quality in Male Broiler Chickens". ‎

We have revised the manuscript based on the reviewers’ comments point-by point ‎and ‎our responses to the reviewer are marked up using the “Track Changes” each ‎specific ‎comment.

We hope the changes made are satisfactory to Your Excellency and to the ‎respected ‎reviewers. ‎

Best regards! ‎

Prof. Ibrahim A. Alhidary

Q1: The current manusript entitled "Effects of Gum Arabic (Acacia Senegal) powder on Characteristics of Carcass, Breast Meat Physical and Quality in Male Broiler Chickens" is about the effect of using gum Arabic in the broiler nutrition. Despite being interesting in the subject, however, it needs improvements in the different sections, especially discussion.

Authors’ Response: Thank you for pointing out that. Done as requested in all manuscript.

Comments in the pdf file

Q2: Lines 2-4: manuscript title.

Authors’ Response: Thanks for your feedback. We revised as required in title.

Q3: Line 17: Please show the P-value.

Authors’ Response: Done as requested in line 18.

Q4: Line 17: Please mention the parameters (e.g., body weight,.....).

Authors’ Response: Done as requested in line 18-19.

Q5: Line18: Please paraphrase this sentence. Difficult to understand.

Authors’ Response: Thanks so much for your notification. We revised as required in line 19.

Q6: Line19: Please report the P-value.

Authors’ Response: Done as requested in line 21.

Q7: Lines 25-26: add

Authors’ Response: Done as requested in line 28-29.

Q8: Line 64: Did you do any analysis to support this?

Authors’ Response: Thank you for your notes. the sentences have been reformulated in all the line that were referred to by you (Line 72-77).

Q9: Line 84: How did you come up with these percentages? Please explain.

Authors’ Response: Thanks so much for your notification. we have explained in line that was referred to by you (Line 94-96).

Q10: In Table 3: what is your explanation for this increase in WL?

Authors’ Response: Done as requested in line 347-350.

Q11: In Table 6: (CP) Please explain about the reasons for this increase in the discussion.

Authors’ Response: Done as requested in lines 401-407.

Q12: In Table 6: (CF) Please explain about the reasons for this decrease in the discussion.

Authors’ Response: Done as requested in lines ‎401-407‎.

Q13: In Table 7: Please discuss in the discussion for the decrease in omega 3 concentration with increase with gum Arabic.

Authors’ Response: there is a need to confirm the mechanism of action of gum ‎Arabic by conducting future studies on characteristics of breast ‎meat quality and fatty acid profile ‎of broiler ‎chickens due to not studies in they. This is what we are working hard in the ongoing studies on the mechanism of action of gum arabic.

But Done as requested in line 415-423. The opinion of the authors, but Please if there are any other opinions, guide us to correct it.

Q14: The discussion is too general. Please be specific. For example, elaborate on the reasons for the improving performance or improving CP and CF in breast meat after using gum Arabic. Discuss little bit on the mechanisms for these happening.

Authors’ Response: Thank you for your interest in these main points. We hope that we have fulfilled the purpose in all discussion.

Q15: Line 361: Please write the full name.

Authors’ Response: Done as requested in line 408.

Q16: Lines 365-366: How do you mention that since no effects was observed on healthy fatty acids like omega 3 and ......

Authors’ Response: Thanks so much for your notification. Done as requested in line 415-419.

Q17: Conclusion. Please paraphrase all the sentences.

Please be specific on your findings.

For example, mention on what parameters were improved by using gum Arabic and which treatment showed the best outcome. Mention on the implications of your study. At the end, give some practical advice for the future of this study.

Authors’ Response: Done as requested in conclusion.

Q18: Line 371: No study was previously conducted to show........

Authors’ Response: the sentences have been reformulated in the line that were referred to by you.

Q19: Line 377: If it was effective on the healthy fatty acid mention it otherwise to delete it.

Authors’ Response: the sentences have been reformulated in the line that were referred to by you.

Q20: The similarity is also 33% which should be less than 20%.

Authors’ Response: Thanks, we revised the whole manuscript carefully to correct any similarity with repeatedly checked by the authors using Plagscan (15% after changes), which is certified by King Saud University. Thank you and appreciate your concern.

Q21: Moderate improvement is needed in English Language.

Authors’ Response: All authors individually revised the whole manuscript carefully to correct any mistakes in the grammar and ‎spelling of writing.

The manuscript has been completely revised, with some language changes made and ‎improved from our point of view, on the other hand, I appreciate your efforts in your ‎valuable comments and question, which gave me the opportunity to improve the ‎throughout manuscript.

Please if there are any opinions, guide us to correct it. ‎

satisfactory to you ‎

Thanks so much for your efforts. Your feedbacks are very valuable and will improve ‎my research skills and biological insight on my future studies.

--

Prof. Ibrahim A. Alhidary

Reviewer 3 Report

Line 42: Please write clearly as you have not used abbreviations before "AGP"

Lines 82-84: How did you determine the gum arabic ratios in the feed?

Line 103: Replace "live weight" with "body weight."

Line 117: The methods in this section are standard methods. Why did you cite yourself and your friends? "calculated [25]., Suliman, G. M.;"

Lines 121-129: The methods in this section are standard methods. Why did you cite yourself and your friends? "Suliman et al. [25].", " Qaid et al. [3].", "Valizadeh et al. [27] and Cázares-Gallegos et al. [28]."

Lines 132-139: The methods in this section are standard methods. Why did you cite yourself and your friends? "Hussein et al. [29] and Adeyemi [30]."

Table 2: The most important factor affecting body weight gain is the chick's initial body weight. In the table, the initial body weight of the chicks belonging to the groups should be given.

Table 3. Why didn't you do lettering even though the difference between groups was statistical for WL.

Line 210: replace "(P =0.0001)" with "P<0.05)"

Line 267: replace "(P =<0.0001)" with "P<0.05)"

Line 210: replace "(P =0.0001)" with "P<0.05)"

Author Response

Date: 20, June, 2022‎

Dear Dr. Reviewer #3‎, ‎

On behalf of my colleagues, we would like to thank you for considering our ‎‎manuscript entitled "Effects of Gum Arabic (Acacia Senegal) powder on Characteristics of Carcass, Breast ‎Meat ‎Physical and Quality in Male Broiler Chickens". ‎

We have revised the manuscript based on the reviewers’ comments point-by point ‎and ‎our responses to the reviewer are marked up using the “Track Changes” each ‎specific ‎comment.

We hope the changes made are satisfactory to Your Excellency and to the ‎respected ‎reviewers. ‎

Best regards! ‎

Prof. Ibrahim A. Alhidary

Q1: Line 42: Please write clearly as you have not used abbreviations before "AGP"

Authors’ Response: Done as requested in line 48.

Q2: Lines 82-84: How did you determine the gum arabic ratios in the feed?

Authors’ Response: Thanks so much for your notification. we have explained in line that was referred to by you (Line 94-96).

Q3: Line 103: Replace "live weight" with "body weight."

Authors’ Response: Done as requested in line 118.

Q4: Line 117: The methods in this section are standard methods. Why did you cite yourself and your friends? "calculated [25]., Suliman, G. M.;"

Authors’ Response: Done as requested in line 133.

Q5: Lines 121-129: The methods in this section are standard methods. Why did you cite yourself and your friends? "Suliman et al. [25].", " Qaid et al. [3].", "Valizadeh et al. [27] and Cázares-Gallegos et al. [28]."

Authors’ Response: Thank you for pointing out that. Done as requested in line 140-144.

Q6: Lines 132-139: The methods in this section are standard methods. Why did you cite yourself and your friends? "Hussein et al. [29] and Adeyemi [30]."

Authors’ Response: Thank you for pointing out that. Done as requested in line 150. and throughout the text (all methods).

Q7: Table 2: The most important factor affecting body weight gain is the chick's initial body weight. In the table, the initial body weight of the chicks belonging to the groups should be given.

Authors’ Response: Thank you very much for your interest in this main point. Done as requested in table 2.

Q8: Table 3. Why didn't you do lettering even though the difference between groups was statistical for WL.

Authors’ Response: Thanks for your feedback. Done as requested in line 214 and Table 3.

Q9: Line 210: replace "(P =0.0001)" with "P<0.05)"

Authors’ Response: Done as requested in line 231.

Q10: Line 267: replace "(P =<0.0001)" with "P<0.05)"

Authors’ Response: Done as requested in line 290.

The manuscript has been completely revised, with some language changes made and ‎improved from our point of view, on the other hand, I appreciate your efforts in your ‎valuable comments and question, which gave me the opportunity to improve the ‎throughout manuscript.

Please if there are any opinions, guide us to correct it. ‎

satisfactory to you ‎

Thanks so much for your efforts. Your feedbacks are very valuable and will improve ‎my research skills and biological insight on my future studies.

--

Prof. Ibrahim A. Alhidary

Round 2

Reviewer 1 Report

The corrections are satisfactory for me and the paper can be published in present form. 

Reviewer 3 Report

Dear Editor, 

The authors have made the minor edits indicated.

Best regards,